# Citizen Science Mosquito Surveillance by Ad Hoc Observation Using the iNaturalist Platform

**DOI:** 10.3390/ijerph19106337

**Published:** 2022-05-23

**Authors:** Larissa Braz Sousa, Stephen Fricker, Cameron E. Webb, Katherine L. Baldock, Craig R. Williams

**Affiliations:** 1UniSA Clinical and Health Sciences, University of South Australia, Adelaide, SA 5000, Australia; larissa.braz_sousa@mymail.unisa.edu.au (L.B.S.); stephen.fricker@unisa.edu.au (S.F.); 2Australian Centre for Precision Health, University of South Australia, Adelaide, SA 5001, Australia; katherine.baldock@unisa.edu.au; 3Medical Entomology, NSW Health Pathology, Westmead, NSW 2145, Australia; cameron.webb@health.nsw.gov.au; 4Sydney Institute for Infectious Diseases, University of Sydney, Sydney, NSW 2006, Australia; 5UniSA Allied Health and Human Performance, University of South Australia, Adelaide, SA 5001, Australia

**Keywords:** citizen science, mosquito, mobile application, public health

## Abstract

Citizen science mosquito surveillance has been growing in recent years due to both increasing concern about mosquito-borne disease and the increasing popularity of citizen science projects globally. Health authorities are recognising the potential importance of citizen science to expanding or enhancing traditional surveillance programs. Different programs have shown success in engaging communities to monitor species of medical importance through low-cost methods. The Mozzie Monitors project was established on iNaturalist—an open citizen science platform that allows participants to upload photos (i.e., observers) and assist identification (i.e., identifiers). This article describes the likelihood of citizen scientists submitting photos of mosquitoes, assesses user submission behaviour, and evaluates public health utility from these citizen science-derived data. From October 2018 to July 2021, the Mozzie Monitors project on iNaturalist received 2118 observations of 57 different species of mosquitoes across Australia. The number of observers in the system increased over time with more than 500 observers and 180 identifiers being active in the project since its establishment. Data showed species bias with large-bodied and colourful mosquitoes being over-represented. Analyses also indicate regional differentiation of mosquito fauna per state, seasonality of activity, and ecological information about mosquitoes. The iNaturalist citizen science platform also allows connectedness, facilitated communication and collaboration between overall users and expert entomologists, of value to medical entomology and mosquito management.

## 1. Introduction

Several on-line platforms, such as Zooniverse, eBird, GLOBE Observer, Mosquito Alert and iNaturalist facilitate citizen science activities and data collection [1,2,3,4,5,6]. The iNaturalist platform (https://www.inaturalist.org, last accessed on 20 March 2022), either as a smartphone application (app) or web-based, is amongst the most commonly used tools used to find and organise biodiversity findings observed by citizen scientists worldwide [7,8]. A joint initiative of the California Academy of Sciences and the National Geographic Society, iNaturalist facilitates an established social network of scientists and citizen scientists who share observations of biodiversity across the globe. As of March 2022, over 2 million people contributed to iNaturalist by sharing their observations (including photographs or audio recordings), and more than 200,000 people assisted with identifying organisms included in those observations. Users can either take part in specific groups of studies or conversations about species identification [4,8]. Several programs that used iNaturalist to identify species and provide a real-time and geographic location were successful in local and international biodiversity monitoring, empowering volunteers with little formal expertise to make observations of a variety of species, with expert verification of data provided via the iNaturalist platform [4,7,9,10,11].

Recent literature has investigated several projects worldwide using iNaturalist to record biodiversity data, focusing on species distribution, conservation studies, migration patterns, distribution of invasive species and community engagement [12,13,14,15]. The Australasian Fishes project on iNaturalist is an example of successful engagement between museum curators, taxonomic professionals, researchers and several citizen scientists that expand scientific knowledge in Australia and New Zealand through data shared on the platform [13]. Recently, the effectiveness of iNaturalist was also demonstrated by researchers in Australia in assessing the impacts of bushfires, with the platform providing scientific data on fire severity and biodiversity response, yielding data relevant to understanding future recovery [16].

The utility and relevance of iNaturalist has mainly been assessed either for biodiversity research or in projects with socio-environmental impact [7,11,17,18]. There is a gap in understanding of how iNaturalist could be used to engage the public in data collection to enhance public health. Citizen science has begun to be incorporated into projects designed to improve the health and wellbeing of the community and resources such as iNaturalist may hold great potential to assist with managing the risks of mosquito-borne disease. It has already been applied to the surveillance of exotic mosquitoes (e.g., *Aedes aegypti*, *Aedes albopictus*, *Aedes japonicus*, and *Aedes koreicus*) in Europe where observations uploaded to iNaturalist identified *Ae. albopictus* as the most abundant species observed in 14 countries, strongly aligned with their known seasonality [19]. There may be great utility in applying iNaturalist to enhance or expand mosquito surveillance programs in other regions of the world.

The use of a citizen science approach in mosquito monitoring has been explored since 2011 [20,21,22]. It has provided opportunities to upscale geographic coverage of the traditional methods and gather real-time information related to human–mosquito encounters [23]. This novel approach to mosquito surveillance has been growing in recent years due to the concern about mosquito-borne diseases spreading and the acknowledgement that it can be financially and operationally challenging for health authorities to sustain wide reaching professional mosquito monitoring programs [23,24]. These citizen science-based programs have shown success in engaging communities to assist monitoring species of medical importance through low-cost methods [2,22,25]. A critical outcome of such programs has been the contribution of citizen scientists to the first detection of invasive species in Europe through the Mosquito Alert phone app in Spain [26].

In Australia, the Mozzie Monitors program (https://mozziemonitors.com/, accessed on 20 March 2022) was established in 2018 and has engaged participants in using a readily available and easily operated mosquito trap (i.e., BG-GAT (Gravid Aedes Trap)) to collect mosquito specimens and then to send digital images to researchers for identification, an approach termed “e-entomology”. The program has demonstrated that this approach can record a similar diversity of mosquitoes when compared to a traditional program in South Australia, costing about 25% of the total annual expenses for a professional mosquito surveillance program in the state [24].

Concomitant with participants operating traps and submitting photos of collected specimens, Mozzie Monitors has been promoting the use of iNaturalist to expand the reach of the program to include uploaded images by citizen scientists not directly involved in the trap program. The number of observations and observers has been increasing every year since the Mozzie Monitors (https://www.inaturalist.org/projects/mozzie-monitors-australia, accessed on 20 March 2022) project page was established in 2018. The total number of mosquitoes species recorded on the Mozzie Monitors project on iNaturalist was greater than those recorded by the mosquito trap component of project [27].

To better understand how iNaturalist could be used for public health purposes, by tracking vector mosquitoes and interactions with humans, we reviewed a suite of attributes of the iNaturalist Mozzie Monitors project, specifically (1) distribution of mosquito fauna across Australia; (2) frequency of use and data sharing; (3) ecological association and species interaction through photos and behaviour description (e.g., mosquito biting, mosquito floral feeding); (4) perception and profile of the iNaturalist users’ network.

## 2. Materials and Methods

### 2.1. Definition of Citizen Science

Although there has been debate regarding the cultural, social, and political acceptability of the term ‘citizen science’, here we consider the term to mean public participation in scientific research [28,29,30]. For this research, the citizen science collaboration presented was open to people of any nationality, socio-cultural and economic background [31,32] participating in the Mozzie Monitors project, as well as the broader iNaturalist community invited to participate in an electronic survey. As an open platform, iNaturalist automatically accepts observations from any user, providing they have a parent or legal guardian’s consent to create an account if under 13. This study analysed observations submitted by all users within Australia to assess the utility of mosquito data, and only general users over 18 were invited to answer the electronic survey.

### 2.2. Establishing the Project on iNaturalist

The project was established in October 2018 (https://www.inaturalist.org/projects/mozzie-monitors-australia?tab=about, accessed on 20 March 2022). The project page automatically collects observations of mosquitoes (Diptera Culicidae) (i.e., observations uploaded by individual users of known or suspected mosquitoes) taken from any state and territory in Australia. Users can become members of the project to receive updates. However, they do not need to be members or upload their observations to the project specifically; observations are automatically linked to the project when lodged. The iNaturalist system then collates observations provided they meet the stated criteria (i.e., mosquitoes in Australia). The project accepts any media (photos or sounds) submitted in any month of the year from any place (including houses, open areas, public and private establishments).

### 2.3. Quantitative Methods in Mosquito Community Composition

We extracted data from the Mozzie Monitors project on iNaturalist (https://www.inaturalist.org/projects/mozzie-monitors-australia, accessed on 20 March 2022) using the ‘Export Observations’ tool. To analyse the regional differentiation of mosquito fauna, seasonality of activity, number of observers in the system and mosquito diversity metrics, we exported all data available on each observation collected within the project. The extraction filters selected are described on Table 1.

The date range for submissions to the project included observations uploaded from 19 October 2018 to 29 July 2021. However, as users can upload old photos to the platform, the date range for the actual observations varied from as early as 22 January 2000 through to 29 July 2021.

To determine whether seasonal fluctuations in mosquito populations and communities could be detected, we analysed observations through time. Only research-grade observations (approximately 56% of the total) were used from within the period October 2018–July 2021 (*n* = 34 months) as the status of these observations on iNaturalist are assessed through the Data Quality Assessment—a summary of an observation’s accuracy, completeness, and suitability for sharing with data partners. Observations are qualified to Research Grade status when they have a date, are georeferenced, have photos or sounds, are not of a captive or cultivated organism, and at least 2/3 of identifiers agree on the species-level identification (or other levels, such as family, genus, etc.). The iNaturalist community makes or validates identifications, including expert mosquito entomologists and taxonomists. Previous research has attested to the accuracy of the iNaturalist crowdsourcing research grade system [33]. However, other authors showed that research grade did not accurately identify termite records [34]. Thus, in the Mozzie Monitors project, an expert medical entomologist validated the identifications (authors S. Fricker and C. Webb).

To determine whether observations showed regional differentiation, a dissimilarity analysis was conducted using state and territory specific observations. User behaviours (frequency and number of observations) were characterised through descriptive metrics. We used the full dataset for these analyses, from 2000 to 2021. Data were exported as a CSV file. Data cleaning and initial descriptive analysis were performed on Microsoft Excel (version 2017, © Microsoft Corperation 2017, Washington, DC, USA). Time series and dissimilarity analysis using non-metric multidimensional scaling tool were run in R Core Team (2020).

### 2.4. iNaturalist User Perceptions

To evaluate the potential contribution of iNaturalist users to mosquito surveillance, a survey was developed to assess their perceptions and motivations. The 22-question survey was administered to well established iNaturalist users (those with over 1000 observations of any taxa, not specifically mosquitoes) to explore their perceptions and likelihood of contributing to mosquito observations on a long-term basis. This threshold was chosen as these users were already highly actively engaged in the platform and we hypothesised that this cohort may provide a valuable resource of experienced users where introduction to, and familiarity with, the iNaturalist platform itself would not be a barrier to participation. The survey consisted of 21 multiple-choice questions and one free-text response question. This survey was approved by the University of South Australia’s Human Research Ethics Committee (Ethics approval (202266)).

The survey included questions related to users’ demographics, the reason for using the iNaturalist platform, favourite taxa or observations, interest in specific taxonomic groups, concern about species that can impact public health, and the likelihood of sharing mosquito observations. Respondents were recruited via the messenger tool on iNaturalist (https://www.inaturalist.org/messages, accessed on 20 March 2022). A standard invitation was sent to users from English speaking countries who had over 1000 observations with exclusion criteria including users with less than 1000 observations, under 18 years old, and non-English speakers, due to unavailability of appropriate translation services on the survey platform used. These criteria were determined to explore the profile of active users and the likelihood of people who are already highly active to start/keep sending observations of mosquitoes for a continued period, thus contributing to mosquito surveillance. Users that replied were asked to send their email addresses to receive the information sheet and a link containing the consent form and survey. The electronic survey was hosted by Research Electronic Data Capture (REDCap), available by the University of South Australia credentials through the Australian Access Federation (AAF). Invitations to participate in this research were sent to 530 iNaturalist users.

## 3. Results

### 3.1. Quantitative Mosquito Observation Data

From October 2018 to July 2021, the Mozzie Monitors project on iNaturalist received a total of 2118 observations representing 57 different species of mosquitoes across Australia. In total, 545 observers and 181 identifiers have contributed to the project since its establishment. The number of submissions varied geographically, with observation hotspots concentrated on the southeast coast (Figure 1). The hotspots of observations were concentrated around the capital cities in each of the states along the southeast coast, including Adelaide in South Australia, Melbourne in Victoria, Sydney in New South Wales and Brisbane on the southeast coast of Queensland. There are also additional hotspots along the east coast between Sydney and Brisbane in association with major urban centres including Newcastle, Port Macquarie and Coffs Harbour on the North Coast of New South Wales.

Photos of all mosquito life stages (i.e., eggs, immature stages, and adults) were received. However, research-grade observations used here (*n* = 1187) were predominantly adult stage (approximately 95%). The adult mosquito observations included photos of both females and males, including specimens that were blood fed, biting, or sitting on the skin (36%), interacting with plants (41%), dead (14%), or resting on a wall or similar surfaces (9%). The number of observations varied per state, with the highest number of observations recorded in South Australia (SA), followed by Queensland (QLD) and New South Wales (NSW). However, the greatest number of species was observed in NSW (Table 2).

A total of 57 mosquito species were recorded with species richness and number of observations varying between states and territories (Table 3). Across these 57 species are known species with distinctly different environmental and climatic associations as well as ecological niches. Species were recorded that are typically associated with diverse habitats, from saltwater to freshwater, and from natural habitats to container-inhabiting species adapted to urban environments. The species also varied in ecological and medical importance. *Aedes notoscriptus* was the most abundant species observed and the only species recorded in all states and territories. There was no record of *Aedes albopictus* in Australia, although the species is known to be established in the Torres Strait [35]. *Aedes aegypti* was only observed in Central and Far North QLD, where it is known to be established, with only five observations.

The dissimilarity analysis shows regional differentiation in the mosquito fauna, with Australian states being characterised by particular species (non-metric fit R2 = 0.995, linear fit R2 = 0.964) (Figure 2). In broad terms, the clustering and distancing of states tracks with climate and ecosystem characteristics.

The non-metric multidimensional scaling (NMDS) assesses information from multiple communities or sites, collapsing them into fewer dimensions. This analysis allows data visualisation and interpretation through ordination methods, measuring community dissimilarities [36].

The 15 most observed species were *Ae. notoscriptus*, *Tx. speciosus*, *Ae. vigilax*, *Ae. camptorhynchus*, *Ae. alboannulatus*, *Ae. alternans*, *Ae. vittiger*, *Cx. quinquefasciatus*, *An. annulipes*, *Cq. xanthogaster*, *Cx. annulirostris*, *Cq. linealis*, *Cx. globocoxitus*, *Cx. sitiens* and *Tp. atripes* (Figure 3). These observations included nuisance biting species (*Ae. notoscriptus*, *Ae. vigilax*, *Ae. alboannulatus*, *Ae. alternans*, *Cx. quinquefasciatus* and *Cq. xanthogaster*) and species related to the transmission, and associated human disease, of mosquito-borne pathogens, especially Ross River and Barmah Forest viruses (*Ae. notoscriptus*, *Ae. vigilax*, *Ae. camptorhynchus,* and *Cx. annulirostris*). *Culex annulirostris*, one of the main nuisance pests in Australia, sits in the 11th position in the top observed species, although it is widespread in the country, except for Tasmania. *Anopheles annulipes* is believed to have played a role in historical transmission of malaria in the country.

Medium to large-bodied mosquitoes with bright colours and distinct patterns were highly observed (*Tx. speciosus*, *Ae. alternans*, *Ae. vittiger*, *Cq. xanthogaster*). Additionally, there was a high frequency of observations of mosquitoes (*Ae. vigilax*, *Tx. speciosus*) resting on plants and/or flowers or engaged in blood-feeding behaviour (Figure 4). A remarkable interaction was observed in South Australia, where a specimen of *Ae. camptorhynchus* was recorded for first time (on iNaturalist) trapped in a carnivorous plant from the genus *Drosera* (Figure 4B).

The number of observations, observers and species of mosquitoes shared on iNaturalist have been increasing since the Mozzie Monitors project establishment (Figure 5). The time series plot shows peaks of observations between September to May, from 2018 to 2021. The highest peak of verified observations (research grade and needing ID), observers and species shared was in April 2021.

The cumulative percentage of observations per user showed a higher frequency of observations concentrated on 23 out of 545 observers, representing 50% of the total mosquito observations on iNaturalist in Australia. Thirty-one users contributed at least 10 observations, whereas 337 users contributed with only one observation from October 2018 to July 2021.

### 3.2. iNaturalist User Perceptions

Invitations to participate in this research were sent to 530 iNaturalist users and perception survey was responded to by 309 users from September 2020 to May 2021.

Responses show demographics, preferences and profile of active iNaturalist users and the likelihood of iNaturalist gathering public health utility data (Table 4). Overall, most users were male, highly educated (with university or any other tertiary degree), and have used iNaturalist for 2 to 5 years. They reported using iNaturalist for different reasons, including learning about the natural world, learning about local species, liking science, liking being a citizen scientist, and protecting biodiversity.

Respondents reported interest in observing plants, fungi, mammals, birds, marine and freshwater fauna, herpetofauna and insects. Indeed, insects, proved to be the most popular taxon amongst users. However, 46% were only moderately interested in observing mosquitoes and the same amount was very interested in observing pollinators. Regarding previous experience, knowledge and perceptions, 65% responded that they had already shared observations of mosquitoes, only 11% had heard about the Mozzie Monitors project, and 66% consider important sharing observations of mosquitoes to learn about local mosquito fauna. Finally, 83% of respondents were moderately to extremely likely to share observations of mosquitoes in future.

## 4. Discussion

For the first time, we report here the utility of iNaturalist as a resource to record the diversity of Australia’s mosquito fauna. This platform has proven useful in engaging communities for gathering scientific data, either for biodiversity research, or in projects with socio-environmental impact (e.g., monitoring bushfire recovery in Australia). However, there is a gap in exploring whether iNaturalist could be used to engage the public in data collection aiming to enhance public health, especially by a better understanding of the relative abundance and distribution of endemic, and potentially, exotic vector species.

Australia has a diverse mosquito fauna with distinct geographic distributions. Under the influence of a changing climate, there is often debate about shifting geographic distributions of mosquitoes, either directly resulting from a change in temperature and rainfall patterns or from human facilitated movement [37]. Our analysis demonstrates that the known regional differences in mosquito fauna are reflected in the observations to iNaturalist from individuals in each state and territory and may, consequently, be useful to identify extended geographic ranges or novel introductions. There were no observations that differed markedly from the known or suspected distributions of the mosquito species in Australia [38]. However, there were some noteworthy differences in observations of key species between states and territories. These differences were somewhat surprising given the known distribution and relative abundance of mosquitoes and the relatively low number of observations. This may be due to a number of factors including the number of active iNaturalist users within regions where these mosquitoes are most active or morphological attributes of the individual mosquito species. For example, *Cx. quinquefasciatus* and *Cx. annulirostris* are known to be highly abundant and widespread throughout the country except for Tasmania, [38] but they were not observed by iNaturalist users in all states and territories. These mosquitoes, especially *Cx. quinquefasciatus*, are often active in urban areas and around human habitation but were perhaps less likely to be photographed due to their non-distinctive appearance. Large mosquitoes with bright colours and notable distinguishing patterns, such as patches and stripes, and males with feathery antennae, tended to be over-represented, as observed for *Tx. speciosus*, *Ae. alternans*, *Ae. vittiger* and *Cq. xanthogaster*. This phenomenon has also been found for other taxa, such as observations of birds on iNaturalist [39], with larger, more conspicuously coloured species observed more often. However, while *Tx. speciosus* was the second most observed species recorded on the project, it was only observed in Queensland and New South Wales and not in the Northern Territory where it is also known to be present [38].

The relative frequency of observations was not necessarily representative of the expected relative abundance of key mosquitoes of pest and public health concern in the environment. Key species known to be nuisance-biting pests or vectors of mosquito-borne pathogens were not necessarily the most commonly observed. The large number of observations of *Ae. notoscriptus*, *Ae. vigilax*, and *Ae. camptorhynchus* was to be expected as these mosquitoes are either active within suburban areas or exceptionally abundant. However, other species such as *Cx. annulirostris*, *Ae. procax*, and *Ve. funerea*, that have been associated with either nuisance-biting or arbovirus [38,40,41], were not commonly reported or reported at disproportionately lower frequency than would otherwise be expected. The relatively low number of observations of *Cx. annulirostris* was especially surprising given that this species can be highly abundant and is considered a nuisance-biting pest. As a consequence, it may not be possible to correlate mosquito observations submitted to iNaturalist with pest and public health risks but further research is required to elucidate this relationship and its application to assisting the assessment of mosquito-borne disease risk.

It is important to note some bias in the species richness of mosquito observations by individual iNaturalist users. There is a wide range of expertise among iNaturalist users, both with regard to experience in photographing insects as well as methods of observing mosquitoes. While some photos were taken with smartphones, others were taken with more advanced cameras and lens better suited to macrophotography. The differences in quality of photograph will play an important role in determining the ability to confidently identify the specimen. The entomological experience of the photographer, irrespective of quality of photographic equipment, will also determine the likelihood that the specimens will be identified given that key characteristics of mosquitoes may not necessarily be clearly evident in some photographs due to perspective. While many observations uploaded to iNaturalist were of serendipitous encounters with mosquitoes, others were the results of specific efforts (e.g., mosquito trapping) to document mosquitoes by professional entomologists (e.g., formal mosquito surveillance programs; citizen science projects). As a consequence, observations of some mosquito species (e.g., *Ae. wattensis*, *Cx. edwardsi*) through these efforts should not be expected to be replicated through casual observations by the general public.

Observations on the platform demonstrate a seasonal pattern, utility for identifying human-mosquito encounters (especially for vector species), and ecological associations with particular habitats and plants. The photos shared could also allow for studies of floral visitation, given that little is known about the role of mosquitoes as pollinators [42]. Similarly, anthropophilic behaviour of lesser observed mosquitoes may provide useful insights into their potential role in pathogen transmission or pest impacts. The highest number of observations and species reported on the platform match with the known seasonal population dynamics of mosquitoes in Australia, also reported in previous studies [43,44,45,46]. The peak of observations in April could be related to the seasonality of rainfall on the eastern coast in Australia, increase of the platform popularity and the ‘Mozzie Month’ campaign that took place between late February and March in 2021 [47], where users were invited to share their observations of mosquitoes (https://www.inaturalist.org/projects/mozzie-monitors-australia/journal/46613-mozzie-month-challenge, accessed on 20 March 2022). While participants were not specifically asked about whether publicity surrounding ‘Mozzie Month’ influenced their likelihood of submitting observations of mosquitoes, it highlights that specific events of this nature may have potential to increase the number of individuals making observations of mosquitoes or their frequency of observations.

iNaturalist popularity has been increasing, as it is becoming more well known among people interested in biodiversity, researchers and citizen science facilitators. Bioblitz events (e.g., https://www.inaturalist.org/projects/great-southern-bioblitz-2021-umbrella, accessed on 20 March 2022) can also play a role in the increased participation on the platform, as could be observed in Australia in 2020 after the Great Southern Bioblitz when the observations on iNaturalist surpassed 100,000 in a single month for the first time [13]. This raised participation on the platform is also observed in the increasing number of observers on the Mozzie Monitors project. Expanding participation could lead to sustainable engagement in the Mozzie Monitors projects, as was affirmed by 83% of the respondents being moderately to extremely likely to share observations of mosquitoes, and at least 65% of them had already shared observations of this taxon.

Caution is advised when extrapolating the data available on iNaturalist to public health risks. As has been the case with ecological and biodiversity assessments based on iNaturalist observations [48], an acknowledgement of bias is required for a better understanding of the potential application to mosquitoes and mosquito-borne disease. To explore the likelihood of users sharing observations of species related to public health (e.g., mosquitoes), we were able to conduct an assessment of active users’ profile in the platform. Similarly to previous studies on citizen science [49,50,51], users on the iNaturalist platform reported being predominantly male, middle-aged, highly educated and with a strong interest in science. However, they differed from other citizen science studies where female participation was higher [24,52,53]. Research has shown that, depending on the nature of the citizen science activity (whether it is an environmental-oriented recreational hobby, or competitively-driven), it could attract different segments of the population, including males, females, youths, and older adults [54]. As previously observed for the Mozzie Monitors program using the BG-GAT trap, the first demographics assessment showed that the majority of participants were female [24]. Thus, campaigns and recruitment that focus on the public health importance of mosquito observations sharing on iNaturalist, as well as mosquito diversity monitoring, could increase the likelihood of attracting males and females from different ages to participate in the Mozzie Monitors program.

Most respondents showed higher interest in learning about biodiversity than being aware of disease risks or preventing mosquito nuisance. This shows that the profile of most active users in the platform is more biodiversity-oriented, and this interest could be linked to mosquito species monitoring. Indeed, learning about species that occur in their local areas could lead to long-term awareness about these species’ ecological and medical importance.

Although data collected by Mozzie Monitors on iNaturalist has potential for the study of mosquito community composition in Australia, some challenges were identified, such as engaging new users and expanding participation between the established network in order not to over-rely on a few observers, as shown in the user frequency histogram where only 4% of users contributed to 50% of all data shared. Increased participation and users’ retention could improve the identification tool accuracy for mosquitoes in the platform and upscale the geographic coverage of citizen science mosquito surveillance in Australia. Encouraging professional entomologists to join in the platform and assist with identifications could enhance the connectivity between mosquito observers on iNaturalist and mosquito experts, and improve the identification tools based on artificial intelligence and crowdsourcing.

Although many mosquito surveillance citizen science programs have their own customised apps, the use of iNaturalist to monitor mosquitoes could provide a scalable mechanism to engage people with different skills and interests. Designing, maintaining and updating a custom app also requires extra costs and resources. As the iNaturalist user perceptions questionnaire shows, most active users shared varied interests in wildlife, not limited to mosquitoes or public health. Thus, citizen science participation in mosquito monitoring through the iNaturalist platform could contribute to real-time species reporting and identification and increased connectivity between the general community and researchers.

This study also identified some biases and limitations. Results showed that participation was not evenly spread geographically, with a strong location bias. Observers uploaded substantially more photos around the Australian big cities and capitals. Responses to the electronic survey also showed that most active users were highly educated. Perhaps campaigns and educational workshops on using iNaturalist could encourage more people to contribute to citizen science data collection in areas with low participation. This would also be valid in settings with a higher abundance of mosquito-borne diseases; in that case, researchers should consider ethical concerns, such as the geoprivacy of photos showing human–mosquito interaction to track disease progression. Connectedness between users and researchers could increase public health awareness about species vector status and management.

In terms of privacy and security, users can obscure the GPS information of their photos. As an open platform, iNaturalist shows the geographic location of observations. If people or families feel uncomfortable sharing the species observed in their backyards, they have the option to obscure the location. Obscured observations display a 0.2 × 0.2 degrees rectangular area around the hidden coordinates (~500 km^2^ at the equator). When obscured, researchers can still see the specimen’s overall location (state and or suburb). Researchers can approach the observer via iNaturalist chat to ask for more details if a potential vector species is identified in an obscured area. Additionally, the electronic survey did not explore users’ training or background. Further studies could investigate whether there is a correlation between trained scientists participating in iNaturalist and observations and identification accuracy.

As previously demonstrated, online crowdsourcing information on vector arthropods collected data on the seasonality and distribution of vector species [2,26,55]. The potential of iNaturalist for detecting vector species was also verified in Europe, North America, North Africa and the Middle East [19], and the results indicated that the iNaturalist platform could complement existing vector surveillance data. Citizen science initiatives on mosquito surveillance are also emerging in tropical countries and areas where dengue and malaria are endemic [22,56,57], and ethical issues were discussed in a community-based program in Nicaragua and Mexico [58]. As ethical discussions underlining community-based research have shown, researchers should ensure that individuals and communities have the autonomy to monitor and control the vectors in their properties and have a safe space for a dialogue focusing on mutual respect and community health [58]. Future studies regarding the utility of iNaturalist for public health should consider the ethical issues of vector surveillance in impoverished locations where diseases such as dengue and malaria are a major problem.

Research has shown that iNaturalist users were able to register critically endangered species, as well as extremely rare species and behaviours [13]. Similarly, the Mozzie Monitors project on iNaturalist could provide an early-warning system for detection of invasive species, as in detecting the first record of *Aedes (Downsiomyia) shehzadae* in Queensland [59]. It could also facilitate early detection of growing vector mosquito populations through citizen science.

Finally, it is notable that participation in iNaturalist has been increasing over 2021/2022. In only seven months, from July 2021 to February 2022, 877 new observations were added to the Mozzie Monitors project (an increase of 41% compared to data extracted before July 2021). In addition, 111 new observers contributed to the project, and six new species were added to the list, including *Ae. biocellatus*, *Ae. gahnicola*, *Ae. rupestris*, *Ae. subbasalis*, *Cq. variegata* and *Cx. postspiraculosus*. It is important to note that these species were added by an expert entomologist using surveillance traps. This growing engagement in the platform should be investigated in future studies to explore connectedness between overall users and identifiers and to assess an updated inventory of species list collected through citizen science.

## 5. Conclusions

The Mozzie Monitors project on iNaturalist has shown potential to engage the broader community in mosquito monitoring, collecting data of mosquito community composition from all states in Australia since its establishment. Data showed species bias with large-bodied and colourful mosquitoes being over-represented. Analyses also indicate regional differentiation of mosquito fauna per state, seasonality of activity, an increasing number of observers in the system over time and ecological association.

This citizen science platform also allows connectedness, facilitated communication and collaboration between overall users, and expert entomologists who assist with the identifications, answer questions, and share educative material (including identification keys and guides). The growing network of citizen scientists using the iNaturalist platform to submit observations of mosquitoes corroborates with the user perceptions analyses, where respondents demonstrated being very likely to share observations of mosquitoes.

The engagement on the platform can be complementary to other citizen science mosquito monitoring programs, such as the Mozzie Monitors program with fixed point BG-GAT traps (mozziemonitors.com). Whereas the fixed-point trap-system method yields abundance data of mosquito fauna, the Mozzie Monitors project on iNaturalist yields information about diversity and species distribution. Both methods can serve to upscale geographic coverage and provide real-time information for mosquito monitoring.

Although iNaturalist has been largely explored for its utility in gathering data for biodiversity and conservation studies, we have attested that it could be a potential tool for collecting data on public health and biosecurity concerns. The Mozzie Monitors project on iNaturalist has shown valuable data regarding vector mosquitoes’ occurrence and real-time distribution in Australia. Further research could explore the potential and data utility from the growing network of Mozzie Monitors iNaturalists in Brazil (https://www.inaturalist.org/projects/monitores-mozzie-brasil, accessed on 20 March 2022) and in Southern Africa (https://www.inaturalist.org/projects/mozzie-monitors-southern-africa, accessed on 20 March 2022). The citizen science engagement on Mozzie Monitors projects on iNaturalist in these regions where dengue, malaria and yellow fever are more widespread could help complement traditional surveillance methods, delivering positive public health outcomes.

## Figures and Tables

**Figure 1 ijerph-19-06337-f001:**
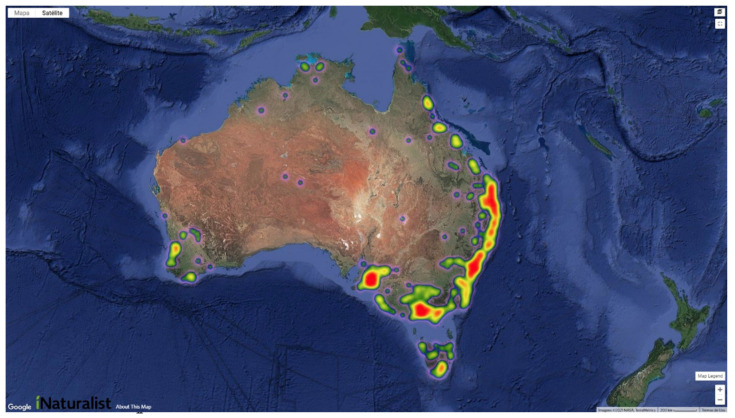
Heatmap showing the distribution of observations on Mozzie Monitors project on iNaturalist, in Australia. The warmer the colour, the more observations submitted in the area. This presented map was adapted from the inbuilt heat map available at https://www.inaturalist.org/observations/map?place_id=any&project_id=mozzie-monitors-australia#5/-28.062/138.359, accessed on 20 March 2022. Map data ©2022 Google. Imagery ©2022 NASA, TerraMetrics.

**Figure 2 ijerph-19-06337-f002:**
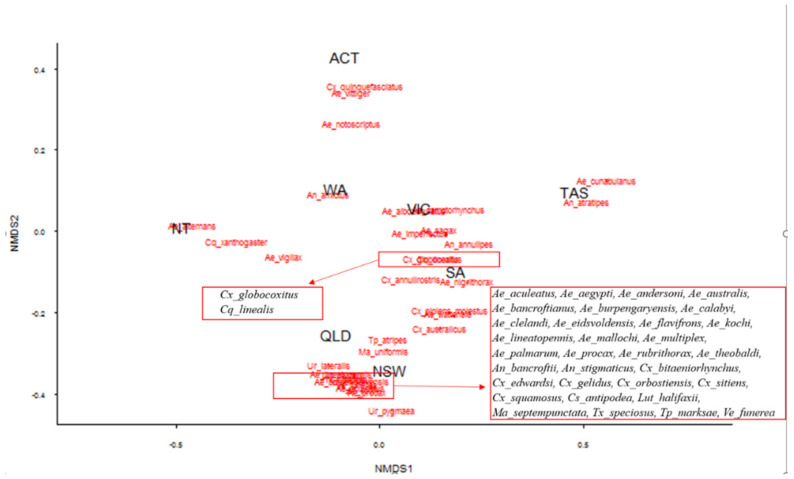
Contribution of mosquito species richness observed per state and territory through a dissimilarity analysis using non-metric multidimensional scaling tool. Axis *x* and *y* show non-metric multidimensional scaling dimensions using Bray-Curtis distances for community-by-site matrix.

**Figure 3 ijerph-19-06337-f003:**
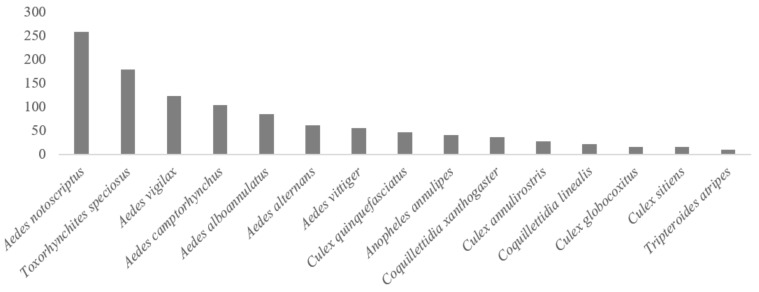
Number of observations of the 15 most observed species of mosquitoes on iNaturalist, in Australia.

**Figure 4 ijerph-19-06337-f004:**
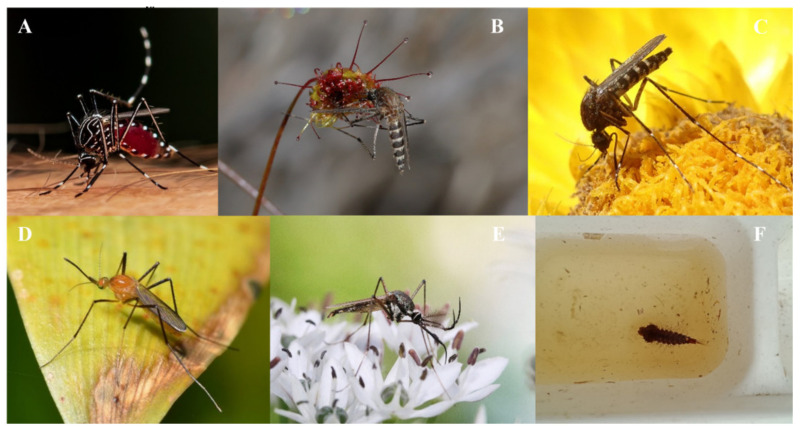
Most common species shared on Mozzie Monitors on iNaturalist with iNaturalist photographer credit. (**A**) *Aedes notoscriptus* © Jacky Lien, (**B**) *Aedes camptorhynchus* © frank_prinz, (**C**) *Aedes vigilax* © Jeannie, (**D**) *Coquillettidia xanthogaster* © Dianne, (**E**) *Toxorhynchites speciosus* © Sylvia Alexander, (**F**) Larva of *Toxorhynchites speciosus* © Gillian Fitzgerald. Photos A-F have Creative Commons license.

**Figure 5 ijerph-19-06337-f005:**
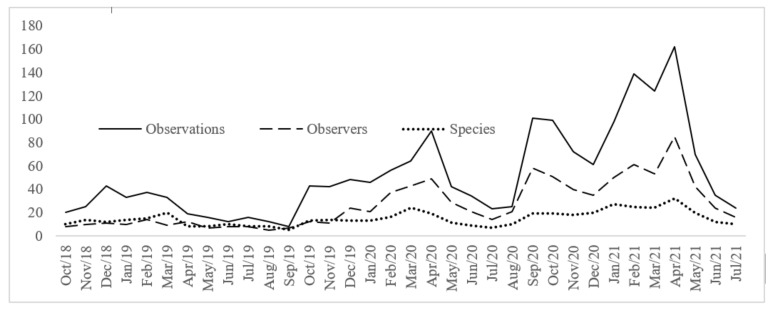
Time series of number of observations, species and observers in Australia, respectively, from October 2018 to July 2021.

**Table 1 ijerph-19-06337-t001:** Extraction filters selected to download mosquito observations from the Mozzie Monitors project on iNaturalist.

Quality Grade	Any (Research Grade, Needs ID, Casual)
Identifications	Any (most agree, some agree, most disagree); Captive/cultivated: any (yes, no)
Project	mozzie-monitors-australia (project settings already include all mosquitoes [Family Culicidae], observed from any state in Australia, uploaded by any users, any quality grade, any media type, and from any date)
Date range	Any
Columns	default selection, containing all data for Basic (ID, user ID, URL, license, time, etc), Geo (all available coordinates) and Taxon.

**Table 2 ijerph-19-06337-t002:** Total number and percentage of species richness observed on the Mozzie Monitors project on iNaturalist per state in Australia, from 2000 to 2021.

	Species	Observations
Australian Capital Territory (ACT) *	3 (5%)	7 (1%)
New South Wales (NSW)	42 (72%)	341 (29%)
Northern Territory (NT)	4 (7%)	34 (3%)
Queensland (QLD)	26 (45%)	308 (26%)
South Australia (SA)	22 (38%)	345 (29%)
Tasmania (TAS)	7 (12%)	11 (1%)
Victoria (VIC)	12 (21%)	125 (11%)
Western Australia (WA)	7 (12%)	16 (1%)
Australia	57 (100%)	1187 (100%)

* The same abbreviations are presented in the next sections of the article.

**Table 3 ijerph-19-06337-t003:** Species richness and number of observations on the Mozzie Monitors project on iNaturalist per state in Australia, from 2000 to 2021.

		ACT	NSW	NT	QLD	SA	TAS	VIC	WA	Australia
1	*Aedes aculeatus*	0	3	0	1	0	0	0	0	4
2	*Aedes aegypti*	0	0	0	5	0	0	0	0	5
3	*Aedes alboannulatus*	0	15	0	1	18	1	46	3	84
4	*Aedes alternans*	0	29	29	0	0	0	2	1	61
5	*Aedes andersoni*	0	0	0	0	0	1	0	0	1
6	*Aedes australis*	0	0	0	0	1	0	0	0	1
7	*Aedes bancroftianus*	0	0	0	0	2	0	0	0	2
8	*Aedes burpengaryensis*	0	1	0	1	0	0	0	0	2
9	*Aedes calabyi*	0	0	0	0	1	0	0	0	1
10	*Aedes camptorhynchus*	0	2	0	0	82	4	14	2	104
11	*Aedes clelandi*	0	0	0	0	2	0	0	0	2
12	*Aedes cunabulanus*	0	0	0	0	0	2	0	0	2
13	*Aedes eidsvoldensis*	0	1	0	0	0	0	0	0	1
14	*Aedes flavifrons*	0	4	0	0	0	0	0	0	4
15	*Aedes imperfectus*	0	1	0	0	0	0	1	0	2
16	*Aedes kochi*	0	4	0	1	0	0	0	0	5
17	*Aedes lineatopennis*	0	1	0	6	0	0	0	0	7
18	*Aedes mallochi*	0	1	0	0	0	0	0	0	1
19	*Aedes multiplex*	0	5	0	0	0	0	0	0	5
20	*Aedes nigrithorax*	0	0	0	0	1	0	0	0	1
21	*Aedes notoscriptus*	5	79	1	48	76	1	42	7	259
22	*Aedes palmarum*	0	1	0	0	0	0	0	0	1
23	*Aedes procax*	0	7	0	1	0	0	0	0	8
24	*Aedes rubrithorax*	0	1	0	0	0	0	0	0	1
25	*Aedes sagax*	0	0	0	0	3	0	1	0	4
26	*Aedes theobaldi*	0	1	0	0	1	0	0	0	2
27	*Aedes vigilax*	0	41	2	26	53	0	0	1	123
28	*Aedes vittiger*	1	12	0	38	0	0	4	0	55
29	*Aedes wattensis*	0	1	0	0	3	0	0	0	4
30	*Anopheles amictus*	0	0	0	1	0	0	0	1	2
31	*Anopheles annulipes*	0	8	0	4	22	1	6	0	41
32	*Anopheles atratipes*	0	1	0	0	0	1	0	0	2
33	*Anopheles bancroftii*	0	0	0	1	0	0	0	0	1
34	*Anopheles stigmaticus*	0	1	0	0	0	0	0	0	1
35	*Coquillettidia linealis*	0	4	0	0	15	0	2	0	21
36	*Coquillettidia xanthogaster*	0	9	2	24	0	0	0	1	36
37	*Culex annulirostris*	0	5	0	9	11	0	3	0	28
38	*Culex australicus*	0	1	0	0	1	0	0	0	2
39	*Culex bitaeniorhynchus*	0	1	0	0	0	0	0	0	1
40	*Culex edwardsi*	0	1	0	0	0	0	0	0	1
41	*Culex gelidus*	0	0	0	1	0	0	0	0	1
42	*Culex globocoxitus*	0	0	0	1	13	0	1	0	15
43	*Culex orbostiensis*	0	2	0	0	0	0	0	0	2
44	*Culex pipiens molestus*	0	1	0	0	4	0	0	0	5
45	*Culex quinquefasciatus*	1	7	0	3	32	0	3	0	46
46	*Culex sitiens*	0	6	0	9	0	0	0	0	15
47	*Culex squamosus*	0	1	0	0	0	0	0	0	1
48	*Culiseta antipodea*	0	1	0	0	0	0	0	0	1
49	*Lutzia halifaxii*	0	1	0	4	0	0	0	0	5
50	*Mansonia septempunctata*	0	0	0	1	0	0	0	0	1
51	*Mansonia uniformis*	0	2	0	3	1	0	0	0	6
52	*Toxorhynchites speciosus*	0	70	0	109	0	0	0	0	179
53	*Tripteroides atripes*	0	1	0	6	3	0	0	0	10
54	*Tripteroides marksae*	0	2	0	0	0	0	0	0	2
55	*Uranotaenia lateralis*	0	0	0	1	0	0	0	0	1
56	*Uranotaenia pygmaea*	0	1	0	0	0	0	0	0	1
57	*Verrallina funerea*	0	5	0	3	0	0	0	0	8

**Table 4 ijerph-19-06337-t004:** Responses summary for questionnaire interrogating iNaturalist user perceptions and motivations regarding mosquito observations.

Demographics						
Gender	Female (27%)	Male (69%)	Other (2%)	Prefer not to say (2%)		
Age	18–30 (16%)	31–40 (15%)	41–50 (21%)	51–60 (19%)	61–70 (22%)	71–80 (7%)
Highest level of education	High school (10%)	University or other tertiary degree (86%)	Other (4%)			
	Less than one year	From 1 to 2 years	From 2 to 5 years	Over 5 years		
How long have you been used the iNaturalist platform?	7%	26%	43%	24%		
	To learn about the natural world	To learn about the species that occur in my local area	I like science	I like to be a citizen scientist	I like protecting the biodiversity	Other
Why do you use iNat?	77%	80%	67%	78%	73%	28%
	Not interested at all	Slightly interested	Moderately interested	Very interested	Extremely interested	
How interested are you in observations of plants?	1%	12%	22%	32%	33%	
How interested are you in observations of fungi?	1%	25%	41%	22%	11%	
How interested are you in observations of mammals?	2%	15%	31%	33%	19%	
How interested are you in observations of birds?	2%	11%	25%	32%	30%	
How interested are you in observations of fish, marine or freshawater fauna?	3%	25%	31%	24%	17%	
How interested are you in observations of amphibians or reptiles?	1%	10%	31%	35%	23%	
How interested are you in observations of insects?	1%	5%	19%	35%	40%	
How interested are you in observations of mosquitoes?	4%	26%	46%	18%	6%	
How interested are you in observations of pollinators?	1%	9%	26%	46%	18%	
	Not worried at all	Slightly worried	Moderately worried	Very worried	Extremely worried	
Are you worried about species related to public health, such as diseases vectors?	16%	37%	36%	9%	3%	
How concerned are you about threats to biodiversity?	0%	1%	8%	34%	57%	
	From home	When I am hiking	When I am walking	From anywhere		
Where do you usually make observations?	7%	9%	8%	76%		
Have you ever shared observations of mosquitoes on iNaturalist?	Yes (65%)	No (35%)				
Have you heard about the Mozzie Monitors project on iNaturalist?	Yes (11%)	No (89%)				
	Not likely at all	Slightly likely	Moderate likely	Very likely	Extremely likely	
How likely would you be to share observations of mosquitoes?	2%	15%	24%	35%	24%	
	To learn about the species that occur in my local area	To be aware of disease risks	To prevent mosquito nuisance	Other		
Why do you think it is important to share observations of mosquitoes?	66%	18%	5%	10%		

## Data Availability

All data on iNaturalist may be found at: https://www.inaturalist.org/projects/mozzie-monitors-australia, accessed on 20 March 2022.

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
