# Peer review of "Citizen Science Mosquito Surveillance by Ad Hoc Observation Using the iNaturalist Platform"

_ijerph, 2022, doi:10.3390/ijerph19106337_

Round 1

Reviewer 1 Report

Dear authors, 

The manuscript as well as your research are very interesting and very well presented. Manuscript is written in a good level of English.  This manuscript is a great contribution to the science but also has a practical use for the  Australian mosquito PCO. 

General comment is that it is not very clear who was sending mosquito reports (photos)? Only selected observers? It was not opened to all citizens of Australia? This part should be clear in Material and Method. Also it is not very clear who identified mosquitos? Please improve these two very important parts of Material and Method. 

Please find below specific comments about MS: 

2.1. Definition of citizen science: This title should be modified to more specific one because these are criteria for this project of Citizen science but not for all types of citizen science projects. Could you please give a information about the age? Only older than 16 or 18 can use it or the project does not have that kind of limits?

L83-L97 If the citizen science of this project works in the same was as presented in L83-L97, than these lines should be part of Material and Method.

L117 order and family should be in small letters (instead of Order and Family), but please delete whole this bracket (of the insect Order Diptera, Family Culicidae) except (Diptera, Culicidae). It is well known that these two categories in bracket mean order and family.

L122 Please specify what criteria. It is not clear if only criterion is that mosquito picture was taken in Australia?

Table 1. Please try to place identification word in one line

L207 Feeding on plants instead of interacting with plants is more appropriate

Table 2 Please write full names of the locations because you have enough space for that and it is more practical so that readers do not search in the text to see what, for example, QLD is.

Table 3. Please give the abbreviations meaning in legend below table

Figure 2. It would be better to give abbreviations of species here to make them more visible. Especially bad is NSW.

L236-L246 Species should be in italic

Figure 4. A) Aedes notoscriptus (c) Jacky Lien. – here this (c) is confusing. Was is supposed to be this ©? Is it necessary full stop after Lien?

L251-L252 In the text authors mentioned: Ae. camptorhynchus was recorded for first time (on iNaturalist) trapped in a carnivorous plant from 252 the genus Drosera (Figure 4). It should be more precise Figure 4B because the figure consists on several different mosquito species.

L262-L272 This is not clear what it is.

Technical side of Table 4 should be improved. Please format the table so that lines and shape are uniformed. This Prefer not to say (2%) could be in a one line. Please make the lines between the questions for better visual presentation of answers.

Example:

Less than one year

From 1 to 2 years

From 2 to 5 years

Over 5 years

could be change to

< 1 year

1-2 years

2-5 years

> 5 years

L327-L328 species should be in italic. Please act accordingly in the whole manuscript.

(See Appendix A) is not given in MS. And why is Appendix A if MS has only one appendix (no Appendix B, C, D)?

Please correct Crall et al in the reference list to the whole list of authors 

Author Response

General comment is that it is not very clear who was sending mosquito reports (photos)? Only selected observers? It was not opened to all citizens of Australia? This part should be clear in Material and Method. Also it is not very clear who identified mosquitos? Please improve these two very important parts of Material and Method. 

Response: Thank you for your comments. Information about observers was further described on the lines 115-119, and about identifiers on the lines 152-159.

2.1. Definition of citizen science: This title should be modified to more specific one because these are criteria for this project of Citizen science but not for all types of citizen science projects. Could you please give information about the age? Only older than 16 or 18 can use it or the project does not have that kind of limits?

Response: Thank you for raising this point. It was addressed on the lines 115-199.

L83-L97 If the citizen science of this project works in the same was as presented in L83-L97, than these lines should be part of Material and Method. 

Response: Lines 83-91 were part of the introduction as they describe another cohort of participants in the Mozzie Monitors program using a passive mosquito trap (BG-GAT). Lines 92-99 provide previous results and preliminary analysis observed when comparing BG-GAT trap and iNaturalist data. Material and Methods describe a new cohort of participants using the citizen science platform iNaturalist.  

L117 order and family should be in small letters (instead of Order and Family), but please delete whole this bracket (of the insect Order Diptera, Family Culicidae) except (Diptera, Culicidae). It is well known that these two categories in bracket mean order and family.

Response: It has been addressed on line 123.

L122 Please specify what criteria. It is not clear if only criterion is that mosquito picture was taken in Australia?

Response: Thank you, it was further detailed on the lines 115-119.

Table 1. Please try to place identification word in one line.

Response: It was addressed on Table 1.

L207 Feeding on plants instead of interacting with plants is more appropriate.

Response: Interesting observation, thank you for sharing that. However, most observations on the platform show mosquitoes resting on plants, not necessarily feeding. As such, we feel that retaining use of the word ‘interacting’ is most accurate.

Table 2 Please write full names of the locations because you have enough space for that and it is more practical so that readers do not search in the text to see what, for example, QLD is.

Response: It was addressed on Table 2.

Table 3. Please give the abbreviations meaning in legend below table.

Response: Abbreviations are presented on Table 2. The same abbreviations are used in all sections of the article.

Figure 2. It would be better to give abbreviations of species here to make them more visible. Especially bad is NSW.

Response: Thank you for this observation. The figure has been revised.

L236-L246 Species should be in italic.

Response: Thank you for picking this up. The original submitted manuscript did not have all species names italicized. I believe something might have happened during the last edition that accidentally formatted the entire text equally. All the species names were properly italicised now.

Figure 4. A) Aedes notoscriptus (c) Jacky Lien. – here this (c) is confusing. Was is supposed to be this ©? Is it necessary full stop after Lien?

Response: Yes, thank you, it is supposed to be the Copyright symbol. It was addressed on the figure’s caption.

L251-L252 In the text authors mentioned: Ae. camptorhynchus was recorded for first time (on iNaturalist) trapped in a carnivorous plant from 252 the genus Drosera (Figure 4). It should be more precise Figure 4B because the figure consists on several different mosquito species.

Response: Thank you, it was addressed on the line 271.

L262-L272 This is not clear what it is.

Response: These lines were accidentally added during the final edition formatting. It was properly removed now.

Technical side of Table 4 should be improved. Please format the table so that lines and shape are uniformed. This Prefer not to say (2%) could be in a one line. Please make the lines between the questions for better visual presentation of answers.

Response: We appreciate these suggestions. We improved the Table 4 presentation.

L327-L328 species should be in italic. Please act accordingly in the whole manuscript.

Response: Species names were revised and italicised.

(See Appendix A) is not given in MS. And why is Appendix A if MS has only one appendix (no Appendix B, C, D)?

Response: The appendix file has been attached.

Please correct Crall et al in the reference list to the whole list of authors.

Response: The automatic reference style used by EndNote (Numbered) shortened all references on the list to et al. Another reference style was selected to show all authors.

Reviewer 2 Report

Thank you for a pleasant read. It was heartwarming to receive an article that was clearly written by authors who have an easy command of the English language. Although there were a few grammatical issues in places, overall the article was well written and easy to follow. I was able to read the entire article in under an hour (the review, of course, took longer!).

That said, there were some unfortunate oversights in places: there were several locations in which template text was left behind (specifically, page 5 lines 211-213 should be removed, as should page 9 lines 262 - 272, and the last sentence on page 15 line 488).

The uplifting tone of the article was appreciated - it’s not often that I see an article about citizen science! The spirit of creating and fostering a community was inherent in the piece, but I have some edits to recommend:

- Additional limitations: privacy and security. If citizen science is to be relied upon for public health data collection, this opens a rather large can of worms. For example, there are many jurisdictions worldwide in which rigorous collection of COVID-19 case data is no longer taking place. It may seem a natural extension to allow individuals to report suspected cases of COVID-19 among their family, friends, colleagues, and neighbours, through an app similar to iNaturalist. In some jurisdictions, this could have life-threatening consequences for all involved (I'm thinking of, for example, the cases of black plague centuries ago in which houses were literally bricked over with inhabitants still inside). While this may not be the case now for COVID-19, public health researchers tend to agree that more pandemics are likely to occur in the future, and depending on their virulence and mortality rate, this may become a major concern. I would argue any human-to-human infectious disease transmission data should be left to professional scientists.

- Additional limitations: ethics. For a vector-borne disease such as malaria, as mentioned in the article, it may be of interest to report human interactions with mosquitoes - however, again, citizen scientists are not subject to the rigorous ethical oversight offered by professional research institutions, and as a result, mistakes as to which mosquito bite resulted in infection could lead to the same dire consequences as listed above. In fact, recommending (lines 486-488) that citizen science be used to monitor dengue, malaria, and yellow fever may have additional repercussions, as these diseases tend to occur in countries with traditionally low average levels of education, thus increasing the likelihood of mistakes and false conclusions leading to potentially disastrous consequences. It’s interesting that later on, the authors show that most users of the iNaturalist app are highly educated, so the opposite side of this argument is that there may not be much uptake of the app in areas with low education, so the point may be moot, but I think it bears expression. Although I appreciate the spirit of this recommendation as an inexpensive way to track disease progression, this recommendation may be a bit misguided.

- In the conclusions, “Southern Africa” is a region, but “Brazil” is a country; both are referred to as countries. Please edit this to list the countries in Africa to which you refer, or to revise “countries” to “regions”, or similar.

- What does BG-GAT stand for?

- Sometimes species are italicized, and sometimes not (e.g., page 8). I’m not familiar with the rule on italicization here, but I wondered if a second look would be helpful.

- The biases were well discussed, but there is one more bias that was hinted at but should be emphasized: there is a strong location bias here. Regions that are highly populated (and, apparently, that have many people with a tertiary level of education) are more likely to have citizen scientists. Thus, when the authors see that the distribution of known mosquito population levels does not match the distribution of reported mosquitoes, one of the reasons may simply be there are no citizen scientists in the area of interest. This is mentioned on lines 325-326 and lines 340-342.

- Table 1: Please edit the size of the first column to allow the word “Identifications” to appear all on one line.

- Page 4: in the definition of “Research Grade” status, it is indicated that at least 2/3 of identifiers must agree on the species-level identification. Is this a common threshold used elsewhere in the entomology literature? Is it common for several expert entomologists to disagree on a species identification? How difficult is it for citizen scientists to make these identifications, and what is the error rate? Without knowing, it is not possible to assess the reliability of the data set and thus impossible to assess the utility.

- Pages 4 and 7: The use of the non-metric multidimensional scaling tool was not clearly discussed. How does this address dissimilarity? What is being scaled, and what does the resulting graph indicate (page 7)? What do NMDS1 and NMDS2 stand for? Are these unitless measures? 

Also, the reference to R Core Team (2020) should be a reference to the specific package used (if one was used), and if an individual package was not used, it should be a reference only to R, not R Core Team. The reference list should include R Core Team, but the in-line reference should only be R, as far as I know. Perhaps the editor could shine a light on this?

- Page 10: It is interesting that most users were highly educated (i.e., at least some tertiary education). The authors define “citizen science” as “public participation in scientific research”, which does not preclude the participation of professional scientists (who are, of course, also members of the public). How many of these participants had a degree in science? It may be that a community is not really being created and fostered at all - the majority of the community may be professional scientists interacting with other professional scientists. It’s not clear that this has been controlled for in the study.

- Page 12: In the first sentence, the authors indicate that they report the utility of iNaturalist; however, as indicated above, without knowing the frequency of incorrect labelling and other data quality issues, it’s not possible to quantify the utility of this data set for public health research. It’s also not possible to quantify the utility of the app in general for public health research without additional measures.

IJERPH requests that reviewers examine the list of references for inappropriate self-citations. Although 8 of the 53 citations were for work completed by at least one of the authors of this article, none of them appeared inappropriate.

Overall, the authors have presented an interesting first foray into discovering the potential for citizen science to inform public health research, but, in my opinion, there are many additional factors that need to be considered before the “utility” of iNaturalist can be elucidated. 

Author Response

Page 5 lines 211-213.

Response: Lines were removed.

Lines 262-272.

Response: Lines were removed.

Additional limitations: privacy and security. 

Response: Additional limitations explored on lines 467-487.   

Additional limitations: ethics.

Response: Additional limitations explored on lines 472-476.   

In the conclusions, “Southern Africa” is a region, but “Brazil” is a country; both are referred to as countries. Please edit this to list the countries in Africa to which you refer, or to revise “countries” to “regions”, or similar. 

Response: Thank you, this has now been fixed.

What does BG-GAT stand for?

Response: Addressed on page 4 line 85.

Sometimes species are italicized, and sometimes not (e.g., page 8). I’m not familiar with the rule on italicization here, but I wondered if a second look would be helpful. 

Response: Non-italicised names were fixed.

- The biases were well discussed, but there is one more bias that was hinted at but should be emphasized: there is a strong location bias here. Regions that are highly populated (and, apparently, that have many people with a tertiary level of education) are more likely to have citizen scientists. Thus, when the authors see that the distribution of known mosquito population levels does not match the distribution of reported mosquitoes, one of the reasons may simply be there are no citizen scientists in the area of interest. This is mentioned on lines 325-326 and lines 340-342

Response: Thank you for this observation. We agree. Additional limitations are now explored on lines 462-467.

Table 1: Please edit the size of the first column to allow the word “Identifications” to appear all on one line. 

Response: Addressed on line 135.

Page 4: in the definition of “Research Grade” status, it is indicated that at least 2/3 of identifiers must agree on the species-level identification. Is this a common threshold used elsewhere in the entomology literature? Is it common for several expert entomologists to disagree on a species identification? How difficult is it for citizen scientists to make these identifications, and what is the error rate? Without knowing, it is not possible to assess the reliability of the data set and thus impossible to assess the utility.

Response: It was addressed on the lines 154-161.

Pages 4 and 7: The use of the non-metric multidimensional scaling tool was not clearly discussed. How does this address dissimilarity? What is being scaled, and what does the resulting graph indicate (page 7)? What do NMDS1 and NMDS2 stand for? Are these unitless measures? 

Response: Thank you for discussing these points. We provided additional information regarding analysis choice and utility on the lines 253-257.

Also, the reference to R Core Team (2020) should be a reference to the specific package used (if one was used), and if an individual package was not used, it should be a reference only to R, not R Core Team. The reference list should include R Core Team, but the in-line reference should only be R, as far as I know. Perhaps the editor could shine a light on this?

Response: The reference for the “vegan” package is provided on the line 258.

Page 10: It is interesting that most users were highly educated (i.e., at least some tertiary education). The authors define “citizen science” as “public participation in scientific research”, which does not preclude the participation of professional scientists (who are, of course, also members of the public). How many of these participants had a degree in science? It may be that a community is not really being created and fostered at all - the majority of the community may be professional scientists interacting with other professional scientists. It’s not clear that this has been controlled for in the study.

Response: It is addressed as another limitation on the lines 484-487.

Page 12: In the first sentence, the authors indicate that they report the utility of iNaturalist; however, as indicated above, without knowing the frequency of incorrect labelling and other data quality issues, it’s not possible to quantify the utility of this data set for public health research. It’s also not possible to quantify the utility of the app in general for public health research without additional measures.

Response: This limitation is indicated on the lines 422-425.

IJERPH requests that reviewers examine the list of references for inappropriate self-citations. Although 8 of the 53 citations were for work completed by at least one of the authors of this article, none of them appeared inappropriate.

Response: Thank you.

Overall, the authors have presented an interesting first foray into discovering the potential for citizen science to inform public health research, but, in my opinion, there are many additional factors that need to be considered before the “utility” of iNaturalist can be elucidated. 

Response: Thank you for your comments. Additional limitations, biases and future directions were added to the discussion section.

Reviewer 3 Report

Overall the manuscript is of excellent quality; well written, innovative and organized well. I recommend this manuscript is published with minor revisions (see below).

As presented Fig 2 is difficult to read. In place of species names, would color coded shapes be more digestible?  It is also not clear what the reader is expected to conclude from Figure 2, is this meant to demonstrate that ID’s were as expected based on known distributions?  Additional context is needed to understand the purpose and outcomes/expectations of this analysis – in both the body of the paper and the abstract.

Lines 262-272 Typo?; Missing subsection header?

The discussion would be more streamlined if authors more clearly defined limitations and recommendations to overcome those limitation and how to best implement programs using iNaturalist/ similar platforms.

Author Response

Figure 2.

Response: Explanation added on lines 252-258.

Lines 262-272.

Response: Lines were removed.

Discussion and limitations.

Response: Limitations explored on lines 467-487.

Round 2

Reviewer 2 Report

Thank you for addressing the points indicated. I think there are some minor changes that would still make this a stronger article:

I still struggle to understand Figure 2. I appreciate that there is significant scaling happening, and it may be difficult to state directly what is on each axis, but I think without clear labels this is not a publication quality image (and it's also blurry). Perhaps a different image would suit.

Lines 422-425 do a good job of illustrating that interest in iNaturalist and similar apps may increase; however, there isn't a clear indication of how useful iNaturalist/similar apps would be for public health, and with the other limitations including ethical concerns, it seems that reporting mosquitos that traditionally carry dengue and other vector-borne diseases may not be best carried out using citizen science. Perhaps the title of the article could be changed - the case for public health utility has not been clearly made, in my opinion.

Author Response

Reviewer comment: Thank you for addressing the points indicated. I think there are some minor changes that would still make this a stronger article:

Response: we appreciate the work of this reviewer in so rapidly and thoroughly considering our manuscript. Without doubt, the MS has improved as a result.

Reviewer comment: I still struggle to understand Figure 2. I appreciate that there is significant scaling happening, and it may be difficult to state directly what is on each axis, but I think without clear labels this is not a publication quality image (and it's also blurry). Perhaps a different image would suit.

Response: thank you for this comment. We are hopeful that with the upload of the final figure file that any resolution issues will be resolved.

Reviewer comment: Lines 422-425 do a good job of illustrating that interest in iNaturalist and similar apps may increase; however, there isn't a clear indication of how useful iNaturalist/similar apps would be for public health, and with the other limitations including ethical concerns, it seems that reporting mosquitos that traditionally carry dengue and other vector-borne diseases may not be best carried out using citizen science. Perhaps the title of the article could be changed - the case for public health utility has not been clearly made, in my opinion.

Response: we note these concerns and have responded in the following ways:

  1. An amended title as per the reviewer’s suggestion, slightly de-emphasising public health utility
  2. Additional text regarding ethical considerations. This text appears as the second round of track changes (in that additional uploaded file) but can also be found in the clean version of the text in the Discussion . In the text we acknowledge the issues regarding ethical considerations, and cite positive examples of how citizen science can play an augmenting role in public health (without replacing the need for oversight by professional public health practitioners).